# Single nucleus sequencing reveals evidence of inter-nucleus recombination in arbuscular mycorrhizal fungi

Eric CH Chen[1], Stephanie Mathieu[1], Anne Hoffrichter[2], Kinga Sedzielewska-Toro[2], Max Peart[1], Adrian Pelin[1], Steve Ndikumana[1], Jeanne Ropars[1†], Steven Dreissig[3], Jorg Fuchs[3], Andreas Brachmann[2], Nicolas Corradi[1]*

[1]Department of Biology, University of Ottawa, Ottawa, Canada; [2]Institute of Genetics, Faculty of Biology, Ludwig Maximilian University of Munich, Munich, Germany; [3]Leibniz Institute of Plant Genetics and Crop Plant Research, Gatersleben, Germany

*For correspondence:
ncorradi@uottawa.ca

Present address: †Ecologie Systematique et Evolution, CNRS, Université Paris Sud, AgroParisTech, Université Paris Saclay, Orsay, France

Competing interests: The authors declare that no competing interests exist.

**Abstract** Eukaryotes thought to have evolved clonally for millions of years are referred to as ancient asexuals. The oldest group among these are the arbuscular mycorrhizal fungi (AMF), which are plant symbionts harboring hundreds of nuclei within one continuous cytoplasm. Some AMF strains (dikaryons) harbor two co-existing nucleotypes but there is no direct evidence that such nuclei recombine in this life-stage, as is expected for sexual fungi. Here, we show that AMF nuclei with distinct genotypes can undergo recombination. Inter-nuclear genetic exchange varies in frequency among strains, and despite recombination all nuclear genomes have an average similarity of at least 99.8%. The present study demonstrates that AMF can generate genetic diversity via meiotic-like processes in the absence of observable mating. The AMF dikaryotic life-stage is a primary source of nuclear variability in these organisms, highlighting its potential for strain enhancement of these symbionts.
DOI: https://doi.org/10.7554/eLife.39813.001

## Introduction

Mating between compatible partners allows organisms to create genetic variation within populations and provides them with opportunities for adaptation to environmental change. In contrast, long-term clonal evolution should lead to the accumulation of deleterious mutations within lineages and, ultimately, to extinction (*Kondrashov, 1988*; *Agrawal, 2001*; *Kaiser and Charlesworth, 2009*; *Keightley and Eyre-Walker, 2000*; *McDonald et al., 2016*). Some eukaryotic groups have seemingly challenged this theory by evolving for millions of years without observable sexual reproduction. Such organisms have been referred to as ancient asexual scandals, and prominent examples of these evolutionary oddities have included the bdelloid rotifers and the arbuscular mycorrhizal fungi (AMF) (*Normark et al., 2003*; *Judson and Normark, 1996*; *Sanders, 1999*). In rotifers, genome diversity can be generated by reshaping chromosomes through horizontal gene transfers (*Flot et al., 2013*; *Debortoli et al., 2016*; *Signorovitch et al., 2016*). Other asexual mechanisms that increase genetic variability in clonal taxa include transpositions, DNA replication or inter- or intra-chromosomal rearrangements (*Faino et al., 2016*; *Croll, 2014*; *Seidl and Thomma, 2014*). Furthermore, the absence of observable mating does not necessarily indicate that sex is absent as, for example, several organisms long thought to be clonal were eventually found to harbor many genomic signatures of sexual reproduction within their genomes. Evidence of such cryptic sexuality includes the presence of meiosis-specific genes, mating-type loci, sex pheromones, and evidence of inter-strain recombination (*Taylor et al., 2015*).

AMF are ecologically successful organisms that have long been characterized as having lacked sexual reproduction for hundreds of millions of years (*Sanders and Croll, 2010*). These fungi form obligate symbioses with the roots of most land plants, and their presence in the soil can improve plant fitness and biodiversity (*James et al., 1998*; *van der Heijden et al., 1998*; *van der Heijden et al., 2015*). Their capacity to colonize early plant lineages such as liverworts and/or hornworts, and the presence of an AMF symbiotic signaling pathway in algal ancestors of land plants, are some indications that AMF-plant associations may have been established when plants colonized land (*Bonfante and Selosse, 2010*; *Delaux et al., 2015*). The AMF mycelium is unique among eukaryotes as it harbors hundreds to thousands of nuclei within one continuous cytoplasm, and there is no known life stage with one or two nuclei. Although sexual reproduction is not formally observed in AMF, these fungi contain many genomic regions with homology to those involved in mating in distant relatives (*Halary et al., 2011*; *Riley and Corradi, 2013*; *Riley et al., 2014*), suggesting that they are able to undergo sexual reproduction. The cellular mechanisms that could trigger sexual processes in AMF have however long been elusive.

Using genome analyses, it was recently shown that isolates of the model AMF *Rhizophagus irregularis* are either homokaryotic – that is co-existing nuclei harbor one putative mating-type (*MAT*)-locus - or dikaryotic – two populations with different nuclear genotypes coexist in the mycelium, each with a divergent *MAT*-locus (*Ropars et al., 2016*). The *MAT*-locus is the genomic region that gives sexual identity to fungal isolates, and isolates of the AMF genus *Rhizophagus* harbor a genomic location with significant similarities to the *MAT*-locus of basidiomycetes (*Ropars et al., 2016*). AMF dikaryons were proposed to originate from hyphal fusion and nuclear exchange (plasmogamy) between compatible homokaryons (*Ropars et al., 2016*). In dikaryotic fungi, compatible nuclei fuse (karyogamy) and undergo recombination, but there is no direct evidence yet that these sexual processes occur in AMF dikaryons. In particular, although inter-isolate recombination has been inferred to exist based on analyses of single loci (*Riley et al., 2014*; *Croll and Sanders, 2009*; *Gandolfi et al., 2003*; *den Bakker et al., 2010*), direct evidence of genetic exchange between nuclei of opposite *MAT*-loci co-existing within the same mycelium, as is expected for fungal mating processes, has never been reported.

## Results

Here, we tested the hypothesis that karyogamy and inter-nuclear recombination occur in the AMF mycelium using a single-nucleus sequencing approach. Sequenced nuclei were isolated from seven isolates of the genus *Rhizophagus,* including three known *Rhizophagus irregularis* dikaryons (A4, A5; SL1 (*Ropars et al., 2016*)), two *R. irregularis* homokaryons (A1, C2 (*Ropars et al., 2016*)), and two closely related species (*Rhizophagus diaphanus* MUCL-43196, *Rhizophagus cerebriforme* DAOM-227022). Genome data from *R. irregularis* SL1 (also known as DAOM-240409), *R. diaphanus* and *R. cerebriforme* represent new additions to public databases from representatives of the Glomeromycotina, and all sequencing data were assembled using SPAdes (*Bankevich et al., 2012*) to facilitate comparisons with published genome data (*Ropars et al., 2016*).

As expected for an AMF dikaryon, the SL1 assembly contains two putative *MAT*-loci, with a genome architecture identical to that of known *R. irregularis* dikaryons (*Figure 1A*). The isolate also harbors the high allele frequency peak at 0.5 characteristic of AMF dikaryons (*Table 1*, *Figure 1B*) (*Ropars et al., 2016*; *Corradi and Brachmann, 2017*). Despite these similarities, the SL1 assembly is more fragmented than those of all other *Rhizophagus* relatives (*Table 1*). Higher fragmentation is also seen in other known dikaryons (*Table 1*; *Ropars et al., 2016*), a feature that probably stems from a higher intra-mycelial genetic diversity in these isolates. Importantly, BUSCO, K-mer graphs and transposable elements (TE) analyses all show that all isolates analysed in this study share highly similar gene repertoires, genome sizes and TE counts (*Table 2*, *Supplementary file 1*). Thus, the higher assembly fragmentation of SL1 may result from other factors such as, for example, the presence of higher heterokaryosis and/or inter-nuclear recombination. In contrast to SL1, the *R. diaphanus* and *R. cerebriforme* assemblies harbor only one *MAT*-locus and show a conventional homokaryotic allele frequency pattern (*Figure 1B*).

To find direct evidence of inter-nuclear recombination in AMF, we obtained and compared partial genome sequences from 86 single nuclei – that is single haploid genotypes – isolated from *R. irregularis* dikaryons (SL1, n = 15; A4, n = 14; A5, n = 8), *R. irregularis* homokaryons (A1, n = 12; C2,

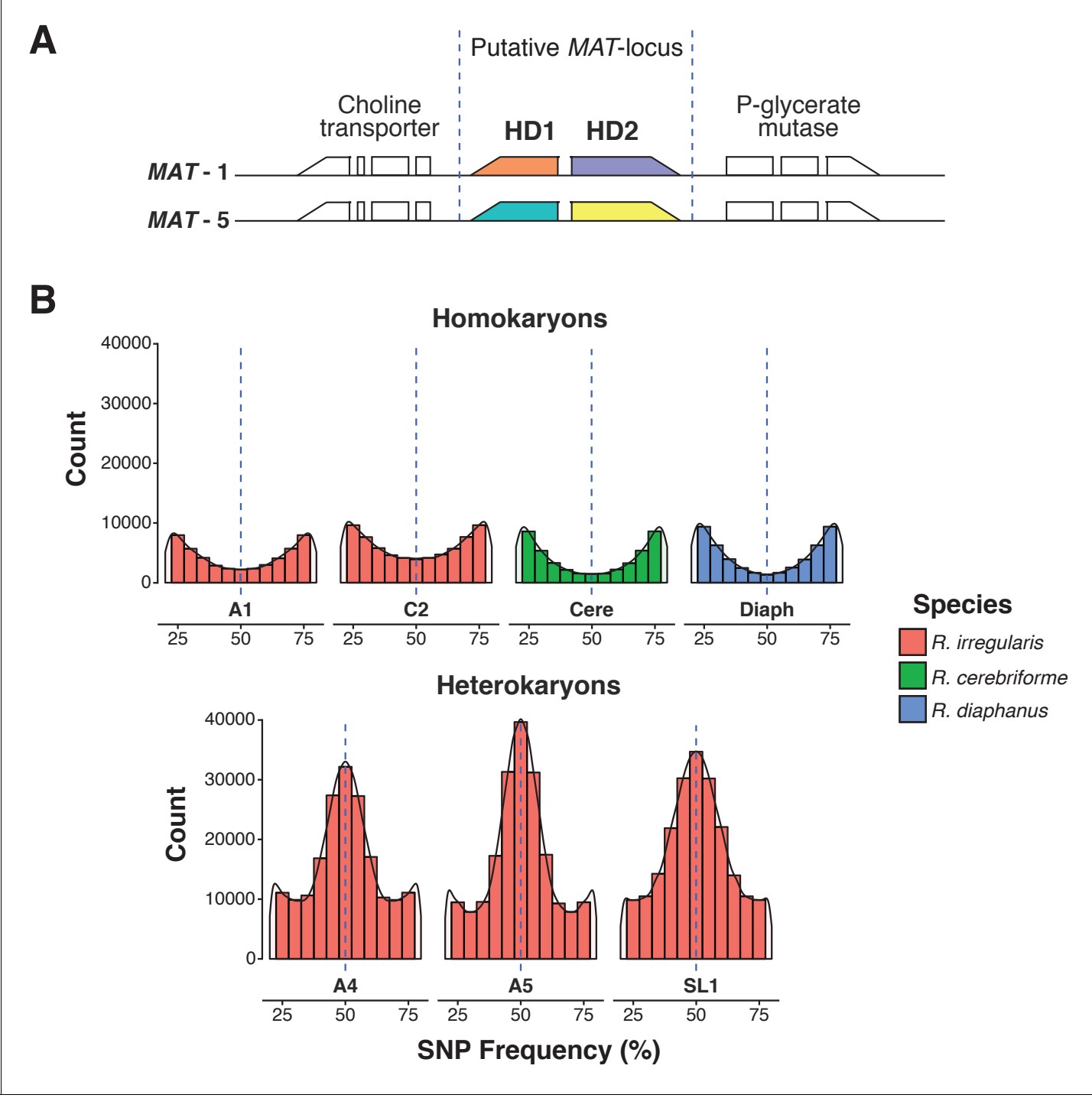

**Figure 1.** A.Predicted *MAT*-locus of *R. irregularis* dikaryon isolate SL1. (**B**) Genome-wide allele frequency of homokaryons and dikaryons. *R. irregularis* SL1 shows the hallmark 50:50 ratio reference to alternate allele frequency of dikaryotic isolates. The *R. irregularis* plots for A1, A4, A5 and C2 represent new analyses of publicly available genome sequence data obtained by (*Ropars et al., 2016*). The homokaryotic isolates *R. irregularis* A1 and C2, and *R. cerebriforme* and *R. diaphanus* are also shown as comparison. Blue dashed line highlights the 0.5 allele frequency.
DOI: https://doi.org/10.7554/eLife.39813.002

**Table 1.** Summary statistics of genomes and nuclei analyzed in this study. * Values from (*Ropars et al., 2016*).

| | *Rhizophagus irregularis* | | | | | *Rhizophagus cerebriforme* | *Rhizophagus diaphanus* |
| | SL1 | A1 | A4 | A5 | C2 | | |
|---|---|---|---|---|---|---|---|
| **A. Genome Assembly Statistics** | | | | | | | |
| Assembly Coverage | 136x | 68x | 95x | 76x | 96x | 150x | 120x |
| Number of Scaffolds | 29,279 | 11,301* | 11,380* | 14,626* | 10,857* | 15,087 | 15,496 |
| Assembly Size (Kb) | 211,501 | 125,869* | 138,327* | 131,461* | 122,873* | 171,896 | 170,781 |
| Assembly SNP/Kb | 0.50 | 0.25 | 0.74 | 0.79 | 0.35 | 0.41 | 0.23 |
| **B. Single Nucleus Statistics** | | | | | | | |
| Number of Nuclei Analyzed | 16.0 | 12.0 | 14.0 | 8.0 | 9.0 | 15.0 | 12.0 |
| Average Assembly Coverage | 14.08% | 57.30% | 46.84% | 58.40% | 61.42% | 22.39% | 11.45% |
| Average Position Depth | 11.8 | 18.1 | 22.1 | 22.7 | 22.9 | 15.4 | 12.3 |
| Number of SNPs Against Reference | | | | | | | |
| – Total SNPs | 346,382 | 241,294 | 423,166 | 291,025 | 251,648 | 262,030 | 101,681 |
| – Average SNPs Per Nuclei - basic filtering | 35,473 | 51,435 | 73,349 | 76,055 | 58,158 | 30,534 | 10,554 |
| – Average Divergence with reference - basic filtering | 0.12% | 0.07% | 0.12% | 0.10% | 0.08% | 0.08% | 0.06% |
| Average - Inter-Nucleus Divergence - basic filtering | 0.38% | 0.13% | 0.24% | 0.16% | 0.14% | 0.24% | 0.21% |

DOI: https://doi.org/10.7554/eLife.39813.003

n = 9), and from the species *R. diaphanus* (n = 12) and *R. cerebriforme* (n = 15). Nuclei were isolated and analysed using a combination of fluorescence-activated cell sorting (FACS), whole genome amplification (WGA) (*Ropars et al., 2016*), and Illumina sequencing. Each nucleus was genotyped by mapping paired-end Illumina reads obtained from WGA-based DNA against their respective genome reference. The combination of PCR-based WGA and Illumina sequencing methods results in variation in average depth position and reference coverage among nuclei and isolates (*Table 1*). Using basic filtering methodologies (see Material and Methods), we find that nuclear genomes diverge from their respective reference from 0.06% for *R. diaphanus*, to a maximum of 0.12% for SL1 and A4 (*Table 1*). Using this method, pairwise nuclear genome comparisons show a high average inter-nuclear similarity ranging from 99.62% in SL1 to a maximum of 99.87% in A1 (*Supplementary file 2*). When present, variability is scattered across the genome (*Supplementary file 3*).

It is noteworthy that most attempts to validate SNP scored using basic filtering methods resulted in the identification of many false positives. Specifically, only 4% (6 out of 148 tested) of SNPs identified using the abovementioned methods were confirmed using PCR and Sanger sequencing procedures performed on the original DNA extracts. In order to maximize the detection of true positives

**Table 2.** BUSCO and K-mer assembly size estimation

| | BUSCO analysis | | K-mer estimation |
| | Genes found | Completeness | Predicted genome size |
|---|---|---|---|
| **A. *R. irregularis*** | | | |
| A1 | 268 | 92.41% | 131.7 Mb |
| C2 | 263 | 90.69% | 148.6 Mb |
| A4 | 262 | 90.34% | 143.5 Mb |
| A5 | 267 | 92.07% | 130.1 Mb |
| SL1 | 265 | 91.38% | 146.5 Mb |
| **B. *R. cerebriforme*** | 267 | 92.07% | 119.7 Mb |
| **C. *R. diaphanus*** | 267 | 92.07% | 121.1 Mb |

DOI: https://doi.org/10.7554/eLife.39813.004

in our study, and thus ensure the detection of *bone-fide* recombinants, we focused our searches for inter-nuclear diversity on the 100 largest scaffolds of each isolate, and scored SNPs only within regions devoid of repeats, manually confirmed to be single copy using reciprocal BLAST procedures, and with frequencies between 0.26–0.74. This conservative approach has likely removed some true positives from our analysis - for example this filtering method resulted in an average 93.8% decrease in total SNP counts (*Supplementary file 4*) - but we believe that this caveat is largely offset by a much higher SNP validation rate, that is 84%, or 30 out of 36 SNPs tested, were validated using Sanger sequencing procedures.

We selected this stringent methodology to genotype all 86 sequenced nuclei and search for evidence of inter-nucleus recombination in all seven isolates. Using this approach, we find that the total number of SNPs detected is, on average, over 30 times larger in dikaryons compared to homokaryotic relatives, with homokaryons harboring only between 9 (*R. diaphanus*) to 131 SNPs (*R. cerebriforme*) along the 100 largest scaffolds (*Supplementary file 4*). Removing the filtering based on allele frequencies resulted in a slightly higher SNP count, but did not alter the overall results – that is dikaryons are still far more variable than homokaryons (*Supplementary file 5*). Consistent with previous findings based on much shorter nuclear regions (*Ropars et al., 2016*), our pairwise sequence comparisons show that nuclei isolated from AMF dikaryons always segregate into two main clades,

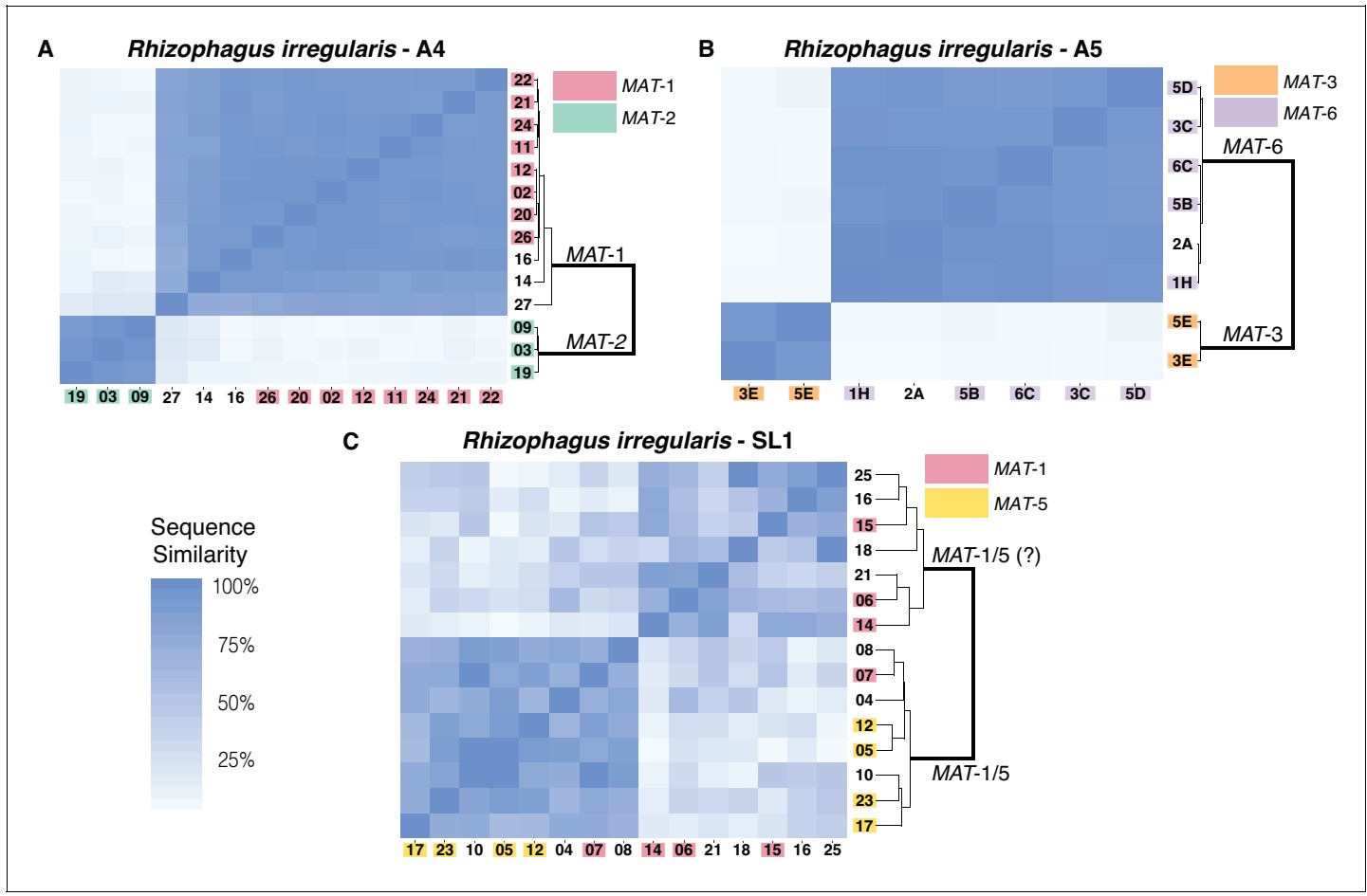

**Figure 2.** Similarity matrix of nuclei isolated from AMF dikaryons. The heat map reflects the level of similarity sequence between nuclei based on the SNPs detected along single copy regions. The number ID of the nuclei with a *MAT*-locus and genotype verified using PCR and Sanger sequencing are shown in coloured boxes. In A4 and A5, clustering of single nuclei correlates well with the *MAT*-locus identity of each nucleus (A, B). In *R. irregularis* SL1, this correlation is not evident, as reflected by the mosaic genetic pattern of single nuclei (C). *R. irregularis* SL1 also shows evidence of inter-nucleus recombination, as evidenced by the high sequence similarity between nuclei with opposite mating-type, for example nucleus 7 (C). Variable regions that could not be assigned to either *MAT*-locus are left with white background.
DOI: https://doi.org/10.7554/eLife.39813.005

depending on their genotype (*Figure 2*), and that each nucleus always harbors one of two co-existing *MAT*-loci.

We find that in A4 and A5, nuclei with the same *MAT*-locus (*MAT*-1 or *MAT*-2 for A4; *MAT*-3 or *MAT*-6 for A5) always harbor highly similar genotypes. This pattern results in a genetic similarity of single nuclei that is based exclusively on the *MAT*-locus identity (*Figure 2A,B*), and we find no evidence that nuclei with opposing *MAT*-loci cluster together in these isolates. In SL1, nuclei also segregate into two dominant clades, a finding consistent with its dikaryotic state. However, obvious links between the *MAT*-locus identity of its nuclei and their genotype are not always evident in this isolate (*Figure 2C*). For example, the nucleus 7 of SL1 harbors the *MAT*-5 locus, but also shows significant sequence similarity with nuclei harboring the *MAT*-1 locus (e.g. nucleus 8 and 12). This finding, which we confirmed using Sanger sequencing approaches (see coloured boxes in *Figure 2*), indicates the presence of inter-nuclear recombination in this isolate, potentially involving the *MAT*-locus. Other nuclei in SL1 show a similar pattern, as evidenced by the distinct mosaic genotypes of this isolate (*Figure 2C*). Overall, the genetic make-up of SL1 contrasts with the clear bi-allelic genetic structure of the isolates A4 and A5 (*Figure 2A,B*), and highlights the higher nuclear genotypic diversity harbored by this isolate.

A detailed look at individual nuclear genotypes highlights evidence of inter-nuclear recombination across the genome (*Figure 3*, *Supplementary file 6*). Examples of recombination tracts include those seen in the SL1 nuclei 14, and 15, which all share the *MAT*-1 locus but can sometimes carry partial genotypes found in co-existing *MAT*-5 nuclei, and *vice versa* (*Figure 3*, *Supplementary file 6*). This pattern of inter-nuclear genetic exchange occurs among many nuclei in this isolate, regardless of their mating-type, and across all of the largest scaffolds in SL1 (*Figure 2*, *Figure 3*, *Supplementary file 6*). In many cases, recombining genotypes encompass hundreds to thousands of base pairs, (*Figure 3*, *Supplementary file 6*). Inter-nuclear exchange between two co-existing genotypes are also seen, although more rarely, in A4 and to a much lesser extent in A5 – for example see scaffold 70 positions 100454 to 100557 (*Figure 3*, *Supplementary file 6*) for potential recombination in A4. In this example, a single recombination event between genotypes harbored by the nuclei 22 (*MAT*-1) and 19 (*MAT*-2) resulted in a genetic exchange involving at least one thousand base pairs, and similar events are found elsewhere in the genome of A4.

We performed identical analyses using different, improved assemblies (e.g. ALLPATHS-LG; (*Butler et al., 2008*); *Supplementary file 7*) and more stringent variant callers (GATK-HaplotypeCaller (*Van der Auwera, 2002*), Mutect2 (*Cibulskis, 2013*); *Supplementary file 8*, *Supplementary file 9*), and found that these did not change the overall findings of our study. Specifically, regardless of the methods used to assemble the genome and call variants, evidence for inter-nuclear recombination is always present and more substantial in SL1 than in in A4 and A5. Accordingly, the genetic relationships among co-existing nuclei are also generally unaffected by changes in both assembly and SNP scoring methods (*Supplementary files 10–12*).

## Discussion

The Meselson effect predicts that the long-term absence of sex and recombination should lead to substantial accumulation of mutations within asexual lineages (*Mark Welch and Meselson, 2000*; *Arkhipova and Meselson, 2005*). Many asexual eukaryotes follow this prediction (*Weir et al., 2016*; *Lovell et al., 2017*; *Neiman et al., 2010*) but our data indicate that AMF may not. Specifically, we find that co-existing nuclei are overall very similar in sequence, showing between 0.001% to maximum of 0.38% average genome divergence, depending on the filters applied to score variation, with the vast majority of variation being in a bi-allelic state. The values obtained using basic filtering methods– that is 0.38% average nuclear genome divergence - are consistent with data obtained by others on single nuclei of the model AMF *R. irregularis* DAOM-197198 (*Lin et al., 2014*), but we also demonstrate that these are largely inflated by false positives. Future work should take this into consideration when assessing AMF genetic diversity using in-silico methodologies. Our study also shows that low nuclear polymorphism is a hallmark of many AMF species and is not only restricted to this *R. irregularis* DAOM-197198. It will be now interesting to see if natural AMF populations show a similarly low nuclear polymorphism.

Our analyses of single nuclei data also supports the hypothesis that essentially all fungal species have found means to recombine (*Nieuwenhuis and James, 2016*). In the case of AMF, our findings

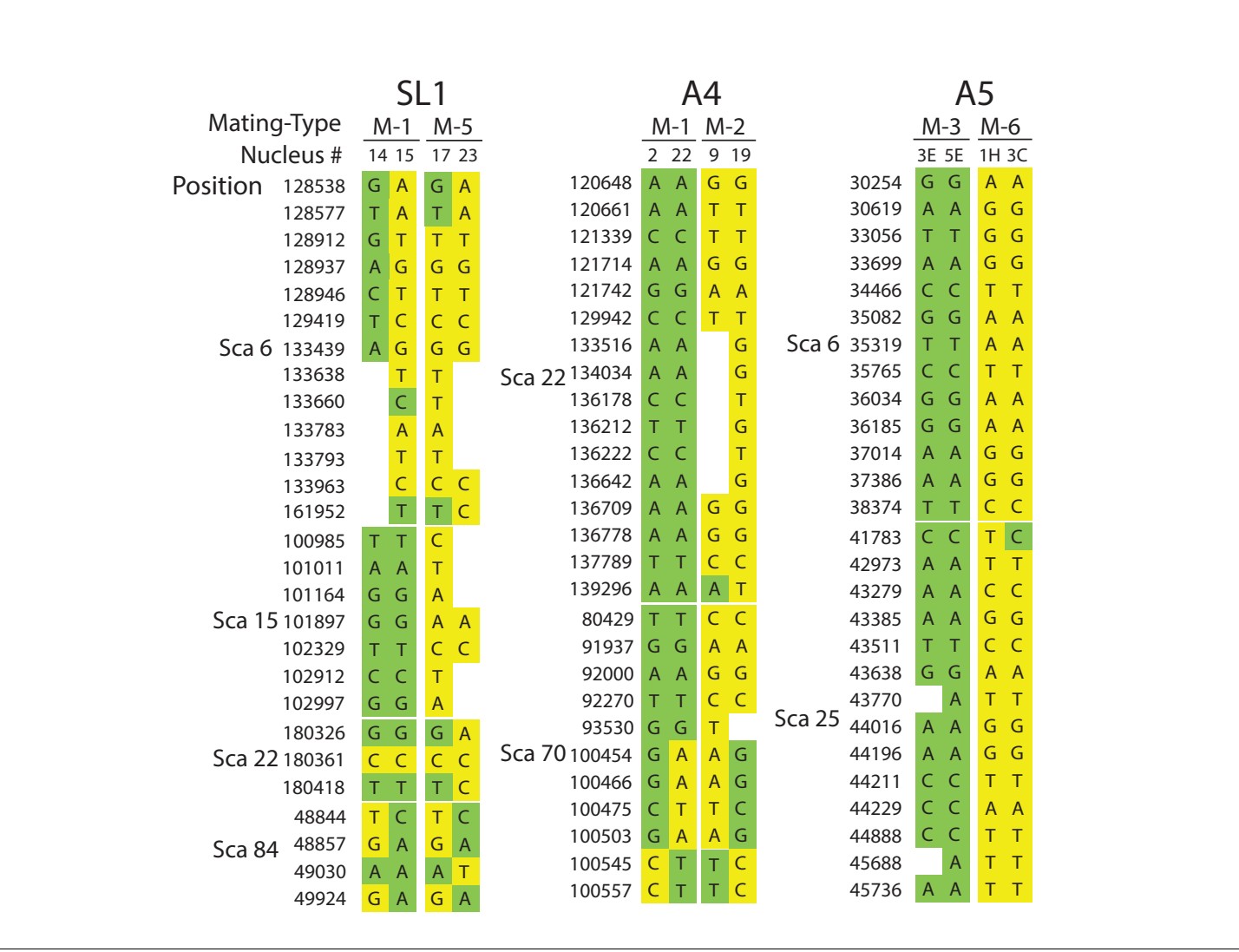

**Figure 3.** Selected examples of genotypes, recombination and inter-nuclear variability observed in the dikaryons SL1, A4 and A5. In SL1, there is no obvious co-linearity between the *MAT*-locus of single nuclei and their genotype. White columns reflect the absence of Illumina sequencing along the nucleus at those homologous positions. The data shown are representative of the genotypes found along the first 100 scaffolds analysed in this study and are based on data available in *Supplementary file 5*. Variations along homologous nucleotide positions are highlighted in yellow or green, depending on the *MAT*-locus which the genotype is expected to associate with. The first and third columns represent, respectively the scaffold and position number where the SNP was scored using stringent filtering methodologies. The second column represents the genotype of the representative genome reference. M: Mating-type.

DOI: https://doi.org/10.7554/eLife.39813.006

build on models previously proposed by Ropars et al. and Corradi and Brachmann (*Ropars et al., 2016*; *Corradi and Brachmann, 2017*), as we found evidence that nuclei with opposite *MAT*-loci can successfully undergo karyogamy in some dikaryotic isolates to generate new nuclear diversity through recombination (*Figure 4*). This discovery is unaffected by the quality of the assembly, and is supported by different methodologies used to score variation. Interestingly, in SL1, the observed inter-nuclear recombination does not appear to result from a single event – that is each nucleus shows differing recombination patterns - an indication that genetic exchanges among nuclei may have occurred independently at different locations within the same AMF multinucleated mycelium (*Figure 4*). In contrast, the few recombination events found in A4 involved exchange of genetic material between two co-existing and highly conserved genotypes.

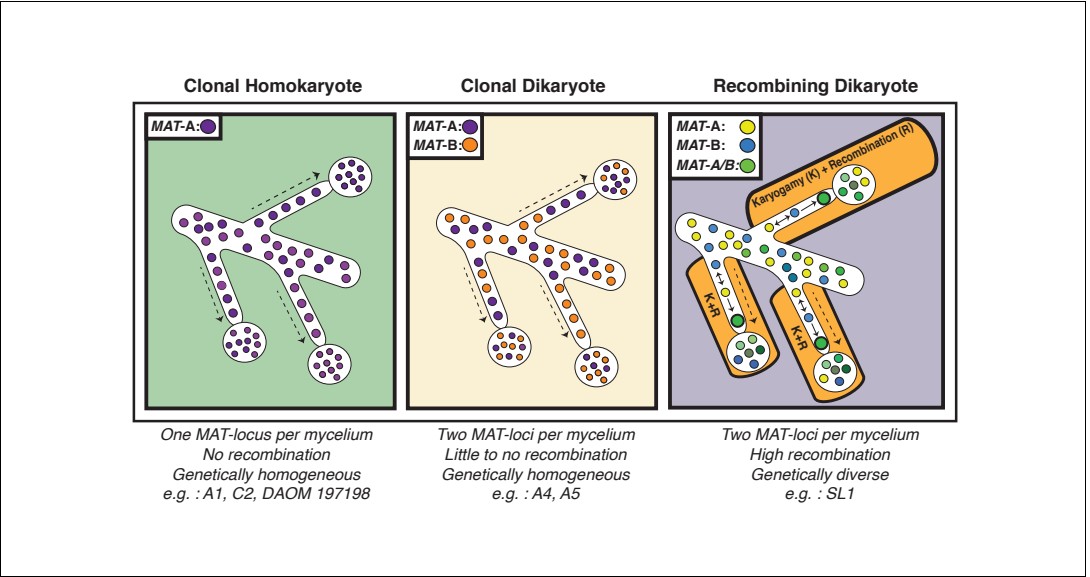

**Figure 4.** Schematic representation of the three genome organizations found to date in the model AMF *Rhizophagus irregularis*. Left: Most AMF analyzed using genome analysis and PCR targeted to the *MAT*-locus have been found to carry nuclei with the same *MAT*-locus. In these isolates, genetic variability is lower than dikaryotic relatives, and recombination is undetectable. Middle: The *R. irregularis* isolates A4 and A5 carry nuclei with two distinct *MAT*-loci. Evidence of recombination is very rare, and two divergent genotypes appear to co-exist in the cytoplasm. Right: In some cases, strains can harbour nuclei with two distinct MAT-loci that undergo frequent karyogamy. The frequency of karyogamy increases nuclear diversity within the mycelium.
DOI: https://doi.org/10.7554/eLife.39813.007

In fungi, diploid nuclei can undergo recombination either through meiosis, in a sexual cycle, or through aneuploidization, in a parasexual cycle involving nuclear fusion followed by random chromosomal loss. Parasexuality is rare in fungi (*Paccola-Meirelles and Azevedo, 1991*; *Rosada et al., 2010*; *Seervai et al., 2013*; *Bennett, 2015*), and this process drives recombination via mitotic events (as opposed to meiosis in a sexual cycle). It also requires the stable division of diploid nuclei for many generations to produce recombination through mitosis (*Clutterbuck, 1996*). To date, flow cytometry data has found that AMF nuclei are haploid (*Ropars et al., 2016*; *Sedzielewska et al., 2011*; *Hosny et al., 1998*) and evidence of widespread diploidy, as evidenced by the mapping of both alternate and reference reads along large portions of genome, could not be found in this study. In contrast, all AMF, including those investigated here, harbor a complete set of meiosis-related genes (*Halary et al., 2011*) and there is recurring evidence that conspecific isolates can exchange and recombine genetic material, presumably through sex (*Riley et al., 2014*; *Croll and Sanders, 2009*; *den Bakker et al., 2010*; *Chen et al., 2018*) . For these reasons, we argue that the internuclear recombination we observed in this study is likely driven by meiotic events - that is conventional fungal sexual processes - although it is possible that parasexuality has also been at play in creating some of the observed nuclear diversity.

What causes recombination rates to vary substantially among the dikaryons SL1, A4 and A5 is presently unclear, as all cultures were established around the same time (*Ropars et al., 2016*; *Jansa et al., 2002*) and there is no available evidence that they were propagated under different in vitro conditions. Perhaps, in isolates such as A4 and A5 the co-existing genotypes are not permissive for generating recombinants (*Idnurm et al., 2015*) - for example the isolates have not yet met the right environmental conditions that trigger karyogamy or lack the mating compatibility necessary to undergo frequent meiosis (*James et al., 2006*; *Heitman et al., 2013*; *Heitman, 2015*). In this case, some of the somatic mutations and recombination found in A4 and A5 could result from exposure to mobile viral genotypes or transposable elements, as there is evidence that these are active in AMF genomes (*Chen et al., 2018*).

In conclusion, our findings demonstrate that arbuscular mycorrhizal fungi can generate genetic diversity via inter-nuclear recombination in the dikaryotic life-stage. This provides a novel understanding of the sexual and genetic potential of these plant symbionts and opens exciting possibilities to enhance their environmental application. In particular, we showed that recombined isolates such as SL1 contain an important source of nuclear diversity, which could lead to genetic enhancement of new AMF strains. Despite evidence of recombination, however, clonality still appears to be the prevalent mode of evolution in lab cultures of these symbionts, including for AMF dikaryons (*Figure 4*). From a conceptual perspective, future work should now aim to identify the mechanisms that trigger inter-nuclear recombination in AMF isolates, and determine how frequent this process is in natural populations of these widespread plant symbionts. It will also be important to establish whether recombination rates vary depending on soil conditions – for example disturbance - or specific plant hosts (*Ritz et al., 2017*), and how this process is reflected at the level of the transcriptome. Within this context, future investigations should also determine if specific recombined genomic regions are linked with an isolate's phenotype (e.g. increased spore production, hyphal density, or mycorrhization rates) or with the fitness of economically important plants and crops.

## Materials and methods

### Single nucleus sorting, WGA, and Illumina sequencing

FACS-based sorting of single nuclei and WGA were performed according to *Ropars et al., 2016* (*Ropars et al., 2016*). DNA concentration of the WGA samples was measured by fluorometric quantification using Qubit dsDNA HS Assay Kit (Thermo Fisher Scientific). Subsequently, sequencing libraries were constructed using Nextera XT DNA Library Preparation Kit (Illumina). Library quality and quantity were checked with a Bioanalyzer and the High Sensitivity DNA Analysis Kit (Agilent Technologies) and a Qubit with the DNA HS Assay Kit (Invitrogen). Samples were sequenced on a HiSeq1500 (Illumina) using 100 bp paired-end sequencing in rapid-run mode at LAFUGA (LMU Genecenter, Munich), resulting in 2,020,347,187 reads. Single nuclei sequencing and genome reads are available in NCBI under the following bioproject: PRJNA477348.

### Genome assembly, read mapping and SNP prediction and filtering

SPAdes v. 3.10 was used to assemble the SL1, *R. diaphanus* (MUCL 43196), and *R. cerebriforme* (DAOM227022) genomes using default parameters (*Bankevich et al., 2012*), and the resulting contigs were then scaffolded using SSPACE (*Boetzer et al., 2011*). Reads from single nucleus sequencing were cleaned using trim_galore with default parameters and were then mapped to the respective genome assembly using BWA-Mem with –M parameter (bwa mem –M) (*Li, 2013*). SPAdes assemblies are publicly available on NCBI (*R. diaphanus*: QZLH00000000, *R. cerebrifrome*: QZLG00000000, SL1 SPAdes: QZCD00000000, and SL1 ALLPATH-LG: QZCC00000000) along with the core meiosis SL1 genes (RAD21/Rec8: MH974797, MND1: MH974798, DMC1: MH974799, Spo11: MH974800, HOP2: MH974801, MSH4: MH974802, MSH5: MH974803).

The whole genome reads are available on NCBI SRA: *R. cerebriforme* (SRR7418134 and SRR7418135), *R. diaphanus* (SRR7418169 and SRR7418170), and *R. irregularis* SL1 (SRR7418171 and SRR7418172). The single nucleus reads are also deposited (*R. cerebriform*: SRR7411799 to SRR7411813, *R. diaphanus*: SRR7410308 to SRR7410319, *R. irregularis* A1: SRR7416439 to SRR7416450, *R. irregularis* A4: SRR7416451 to SRR7416464, *R. irregularis* A5: SRR7416431 to SRR7416438, *R. irregularis* C2: SRR7416465 to SRR7416473, and *R. irregularis* SL1: SRR7411814 to SRR7411829).

Two filtering methods were used to detect variants, which are referred to here as 'basic' or 'stringent'. In the basic filtering procedure, initial variant calling was done using freebayes with the following parameters: -p 1 m 30 K -q 20 C 2; namely a ploidy of one, a minimum quality of mapped reads of 30, a minimum base quality of 20, and a minimum set of reads supporting alternative allele of two (*Garrison and Marth, 2012*). Predicted variant positions were then filtered by keeping only those where reads supporting alternate alleles outnumbered reads supporting the reference allele by a ratio of 10:1. Genotypes of nuclei not supported by the 10:1 read ratio are retained when co-existing nuclei showed the same genotype supported by a 10 to 1 ratio at the same positions.

The strict filtering method was built on top of the basic filtering procedures and includes three additional criteria. The first criterion was the coverage of the reference assembly from the original reads. Specifically, reads were mapped to the reference assembly and only candidate SNP positions with coverage that ranged between 69% and 131% of average coverage along the genome reference (close to 50/50 ratio) were kept. The second criterion was that the remaining candidate SNP positions also needed to keep a proportion of reference allele to alternate allele between 26% to 74% in the SNP calling via original assembly reads. Finally, the third criterion was confirming the single copy nature of the SNP using BLAST procedures. In this case, the 100 bp upstream and downstream regions overlapping the scored SNPs were BLAST against the reference genome. If BLAST results returned more than two good hits (e-value of better than 0.001) this region was considered to be a multi-copy region and thus discarded from downstream analyses.

## Distance matrix comparisons

To detect inter-nuclear sequence divergence, pairwise comparison of SNP data was performed for co-existing nuclei of all isolates. In this analysis, two datasets were created. The first dataset includes only all variable regions, as defined by the total number of filtered SNP positions shared between the nuclei (*Supplementary file 6*). The other dataset includes the number of covered positions shared between the nuclei (*Supplementary file 2*).

## SNP dispersion analysis

Raw SNP predictions from single nucleus reads were cleaned by removing SNP that were longer than length one. Each SNP position was converted into percent of total length of scaffold one and then plotted with an alpha value of 0.2.

## Validation of SNPs and MAT-loci via PCR and Sanger sequencing using DNA from single nuclei

PCR-reactions were run with 0.13 ng/µL to 0.33 ng/µL of single nucleus DNA as a template, 0.5 µM forward and reverse primers each (*Supplementary file 13* and 0.2 mM dNTPs. For reactions performed with Phusion HF Polymerase, 1x Phusion HF Buffer and 0.02 U/µL Phusion HF Polymerase and for reactions with GoTaq Hot-Start Polymerase, 1x Green Buffer, 2.5 mM $MgCl_2$ and 0.025 U/µL GoTaq Hot-Start Polymerase were used. The reactions were performed in a total volume of 15 µL or 30 µL. The amplification was run with a touchdown PCR program: 95°C/2 min - [95°C/30 s; $T_1$/30 s ($T_1$ declining by 0.5°C every cycle) -; 72°C/30 s]x10 - [95°C/30 s; $T_2$/30 s; 72°C/30 s]x35–72°C/5 min. $T_1$ and $T_2$ are specific annealing temperatures for different primer combinations (see *Supplementary file 9*). PCR cleanup was performed with 2.9 U/µL Exonuclease I (New England Biolabs) and 0.14 U/µL Shrimp Alkaline Phosphatase (rSAP, New England Biolabs) at 37°C for 5 min and heat inactivated at 85°C for 10 min. Sanger sequencing reactions were performed at the Genomics Service Unit (LMU).

## Acknowledgements

We thank Timothy Y James and Linda Bonen for their comments on an earlier version of the manuscript. NC is supported by the Discovery program from the Natural Sciences and Engineering Research Council of Canada (NSERC-Discovery), an Early Researcher Award from the Ontario Ministry of Research and Innovation (ER13-09-190), and the ZygoLife project funded by the National Science Foundation (DEB 1441677). AH, KS-T, and AB are supported by the German Research Foundation (DFG grants BR 3527/1–1 and PA 493/11–1).

## Additional information

### Funding

| Funder | Grant reference number | Author |
|---|---|---|
| Natural Sciences and Engineering Research Council of Canada | | Nicolas Corradi |

| Ontario Ministry of Research, Innovation and Science | ER13-09-190 | Nicolas Corradi |
|---|---|---|
| Deutsche Forschungsge-meinschaft | BR 3527/1-1 | Andreas Brachmann |

The funders had no role in study design, data collection and interpretation, or the decision to submit the work for publication.

### Author contributions
Eric CH Chen, Data curation, Formal analysis, Validation, Investigation, Visualization, Methodology, Writing—review and editing; Stephanie Mathieu, Formal analysis, Investigation, Visualization, Methodology, Writing—review and editing; Anne Hoffrichter, Resources, Validation, Investigation, Methodology, Writing—review and editing; Kinga Sedzielewska-Toro, Steven Dreissig, Jorg Fuchs, Resources, Methodology; Max Peart, Resources, Formal analysis, Validation, Methodology; Adrian Pelin, Data curation, Formal analysis, Validation; Steve Ndikumana, Data curation, Formal analysis; Jeanne Ropars, Conceptualization, Writing—review and editing; Andreas Brachmann, Conceptualization, Resources, Formal analysis, Supervision, Funding acquisition, Investigation, Methodology, Writing—review and editing; Nicolas Corradi, Conceptualization, Resources, Data curation, Formal analysis, Supervision, Funding acquisition, Validation, Investigation, Methodology, Writing—original draft, Project administration, Writing—review and editing

### Author ORCIDs
Steven Dreissig http://orcid.org/0000-0002-4766-9698
Nicolas Corradi http://orcid.org/0000-0002-7932-7932

### Decision letter and Author response
Decision letter https://doi.org/10.7554/eLife.39813.057
Author response https://doi.org/10.7554/eLife.39813.058

## Additional files

### Supplementary files
• Supplementary file 1. Number of predicted transposable elements among *R. irregularis* isolates and both versions of SL1 assembly
DOI: https://doi.org/10.7554/eLife.39813.008

• Supplementary file 2. Number of positions shared and total difference between nuclei. Green and yellow are used to distinguish nuclei with a PCR proven MAT-locus. Heat map goes from 0% similarity with orange to 100% similarity with yellow. The pairwise differences between nuclei are small, as expected of nuclei from the same individual.
DOI: https://doi.org/10.7554/eLife.39813.009

• Supplementary file 3. Distribution of SNP (basic filter) along the first scaffold in isolates analyzed. Nuclei that have poor coverage, such as nuclei 13 from A4, were removed. SNPs are distributed fairly evenly across the scaffold although there are apparent hotspots. This may be due to amplification biases.
DOI: https://doi.org/10.7554/eLife.39813.010

• Supplementary file 4. Number of SNP positions after basic and strict filtering. Based on the first 100 scaffolds.
DOI: https://doi.org/10.7554/eLife.39813.011

• Supplementary file 5. Number of SNP positions after strict filtering without taking into account allele frequencies. Based on the first 100 scaffolds.
DOI: https://doi.org/10.7554/eLife.39813.012

• Supplementary file 6. Genotype of variable regions in dikaryons in the first 100 scaffolds. This is the complete list of SNPs used to compute the similarity matrix for *Figure 3*. Nuclei with a PCR proven *MAT*-locus are coloured. Green and yellow are used to highlight the two different genotypes present.

DOI: https://doi.org/10.7554/eLife.39813.013

• Supplementary file 7 Genotype of variable regions in SL1 assembled by ALLPATH-LG in the first 100 scaffolds. This is the complete list of SNPs used to compute the SL1 similarity matrix for *Supplementary file 10* Nuclei with a PCR proven *MAT*-locus are coloured. Green and yellow are used to highlight the two different genotypes present. (A) Variable regions called by freebayes. (B) Variable regions called by HaplotypeCaller. (C) Variable regions called by Mutect2.
DOI: https://doi.org/10.7554/eLife.39813.014

• Supplementary file 8. Genotype of variable regions called by GATK HaplotypeCaller in dikaryons in the first 100 scaffolds. This is the complete list of SNPs used to compute the similarity matrix for *Supplementary file 11*. Nuclei with a PCR proven *MAT*-locus are coloured. Green and yellow are used to highlight the two different genotypes present. (A) *R. irregularis* A4. (B) *R. irregularis* A5. (C) *R. irregularis* SL1.
DOI: https://doi.org/10.7554/eLife.39813.015

• Supplementary file 9. Genotype of variable regions called by GATK Mutect2 in dikaryons in the first 100 scaffold. This is the complete list of SNPs used to compute similarity matrix for *Supplementary file 13*. Nuclei with a PCR proven *MAT*-locus are coloured. Green and yellow are used to highlight the two different genotypes present. (A) *R. irregularis* A4. (B) *R. irregularis* A5. (C) *R. irregularis* SL1.
DOI: https://doi.org/10.7554/eLife.39813.016

• Supplementary file 10. Similarity matrix of nuclei isolated from AMF dikaryons. Same as *Figure 2* but instead of using SPAdes version of SL1 assembly, the ALLPATH-LG version of SL1 assembly is used. The number ID of the nuclei with a *MAT*-locus and genotype verified using PCR and Sanger sequencing are shown in coloured boxes. The patterns remain the same, with A4 and A5 having clear segregation of nuclei based on *MAT*-locus with SL1 showing a much more mosaic pattern.
DOI: https://doi.org/10.7554/eLife.39813.017

• Supplementary file 11. Similarity matrix of nuclei isolated from AMF dikaryons. Same as *Figure 2* but we used GATK's HaplotypeCaller instead of freebayes for SNP calling. The number ID of the nuclei with a *MAT*-locus and genotype that were verified using PCR and Sanger sequencing are shown in coloured boxes. The patterns remain the same, with A4 and A5 having clear segregation of nuclei based on *MAT*-locus with SL1 showing a much more mosaic pattern.
DOI: https://doi.org/10.7554/eLife.39813.018

• Supplementary file 12. Similarity matrix of nuclei isolated from AMF dikaryons. Same as *Figure 2* but we used GATK's Mutect2 instead of freebayes for SNP calling. The number ID of the nuclei with a *MAT*-locus and genotype that were verified using PCR and Sanger sequencing are shown in coloured boxes. The patterns remain the same, with A4 and A5 having clear segregation of nuclei based on *MAT*-locus with SL1 showing a much more mosaic pattern.
DOI: https://doi.org/10.7554/eLife.39813.019

• Supplementary file 13. Primer sequences and combinations for SNP validation of single nucleus DNA.
DOI: https://doi.org/10.7554/eLife.39813.020

• Transparent reporting form
DOI: https://doi.org/10.7554/eLife.39813.021

## Data availability

Single nuclei sequencing and genome reads are available in Genbank under the following bioproject: PRJNA477348. SPAdes assemblies are publicly available on NCBI (R. diaphanus: QZLH00000000, R. cerebriforme: QZLG00000000, SL1 SPAdes: QZCD00000000, and SL1 ALLPATH-LG: QZCC00000000) along with the core meiosis SL1 genes (RAD21/Rec8: MH974797, MND1: MH974798, DMC1: MH974799, Spo11: MH974800, HOP2: MH974801, MSH4: MH974802, MSH5: MH974803). The whole genome reads are available on NCBI SRA: R. cerebriforme (SRR7418134 and SRR7418135), R. diaphanus (SRR7418169 and SRR7418170), and R. irregularis SL1 (SRR7418171 and SRR7418172). The single nucleus reads are also deposited (R. cerebriforme: SRR7411799 to SRR7411813, R. diaphanus: SRR7410308 to SRR7410319, R. irregularis A1: SRR7416439 to SRR7416450, R. irregularis A4: SRR7416451 to SRR7416464, R. irregularis A5: SRR7416431 to

SRR7416438, R. irregularis C2: SRR7416465 to SRR7416473, and R. irregularis SL1: SRR7411814 to SRR7411829).

The following datasets were generated:

| Author(s) | Year | Dataset title | Dataset URL | Database and Identifier |
|---|---|---|---|---|
| Chen ECH, Mathieu S, Hoffrichter A, Sedzielewska-Toro K, Peart M, Pelin A, Ndikumana S, Ropars J, Dreissig S, Fuchs J, Brachmann A, Corradi N | 2018 | Single nuclei sequencing and genome reads from | https://www.ncbi.nlm. nih.gov/bioproject/ PRJNA477348 | NCBI BioProject, PRJNA477348 |
| Chen ECH, Mathieu S, Hoffrichter A, Sedzielewska-Toro K, Peart M, Pelin A, Ndikumana S, Ropars J, Dreissig S, Fuchs J, Brachmann A, Corradi N | 2018 | SPAdes assemblies from | https://www.ncbi.nlm. nih.gov/nuccore/ 1520279022 | NCBI Nucleotide, QZLH00000000 |
| Chen ECH, Mathieu S, Hoffrichter A, Sedzielewska-Toro K, Peart M, Pelin A, Ndikumana S, Ropars J, Dreissig S, Fuchs J, Brachmann A, Corradi N | 2018 | SPAdes assemblies from | https://www.ncbi.nlm. nih.gov/nuccore/ QZLG00000000.1/ | NCBI Nucleotide, QZLG00000000 |
| Chen ECH, Mathieu S, Hoffrichter A, Sedzielewska-Toro K, Peart M, Pelin A, Ndikumana S, Ropars J, Dreissig S, Fuchs J, Brachmann A, Corradi N | 2018 | SPAdes assemblies from | https://www.ncbi.nlm. nih.gov/nuccore/ QZCD00000000.1/ | NCBI Nucleotide, QZCD00000000 |
| Chen ECH, Mathieu S, Hoffrichter A, Sedzielewska-Toro K, Peart M, Pelin A, Ndikumana S, Ropars J, Dreissig S, Fuchs J, Brachmann A, Corradi N | 2018 | SPAdes assemblies from | https://www.ncbi.nlm. nih.gov/nuccore/ QZCC00000000 | NCBI Nucleotide, QZCC00000000 |
| Chen ECH, Mathieu S, Hoffrichter A, Sedzielewska-Toro K, Peart M, Pelin A, Ndikumana S, Ropars J, Dreissig S, Fuchs J, Brachmann A, Corradi N | 2018 | Core meiosis SL1 genes from | https://www.ncbi.nlm. nih.gov/nuccore/ MH974797.1/ | NCBI Nucleotide, MH974797 |
| Chen ECH, Mathieu S, Hoffrichter A, Sedzielewska-Toro K, Peart M, Pelin A, Ndikumana S, Ropars J, Dreissig S, Fuchs J, Brachmann A, Corradi N | 2018 | Core meiosis SL1 genes from | https://www.ncbi.nlm. nih.gov/nuccore/ MH974798 | NCBI Nucleotide, MH974798 |
| Chen ECH, Mathieu S, Hoffrichter A, Sedzielewska-Toro K, Peart M, Pelin A, Ndikumana S, Ro- | 2018 | Core meiosis SL1 genes from | https://www.ncbi.nlm. nih.gov/nuccore/ MH974799 | NCBI Nucleotide, MH974799 |

| | | | | |
|---|---|---|---|---|
| pars J, Dreissig S, Fuchs J, Brachmann A, Corradi N | | | | |
| Chen ECH, Mathieu S, Hoffrichter A, Sedzielewska-Toro K, Peart M, Pelin A, Ndikumana S, Ropars J, Dreissig S, Fuchs J, Brachmann A, Corradi N | 2018 | Core meiosis SL1 genes from | https://www.ncbi.nlm.nih.gov/nuccore/MH974800 | NCBI Nucleotide, MH974800 |
| Chen ECH, Mathieu S, Hoffrichter A, Sedzielewska-Toro K, Peart M, Pelin A, Ndikumana S, Ropars J, Dreissig S, Fuchs J, Brachmann A, Corradi N | 2018 | Core meiosis SL1 genes from | https://www.ncbi.nlm.nih.gov/nuccore/MH974801 | NCBI Nucleotide, MH974801 |
| Chen ECH, Mathieu S, Hoffrichter A, Sedzielewska-Toro K, Peart M, Pelin A, Ndikumana S, Ropars J, Dreissig S, Fuchs J, Brachmann A, Corradi N | 2018 | Core meiosis SL1 genes from | https://www.ncbi.nlm.nih.gov/nuccore/MH974802 | NCBI Nucleotide, MH974802 |
| Chen ECH, Mathieu S, Hoffrichter A, Sedzielewska-Toro K, Peart M, Pelin A, Ndikumana S, Ropars J, Dreissig S, Fuchs J, Brachmann A, Corradi N | 2018 | Core meiosis SL1 genes from | https://www.ncbi.nlm.nih.gov/nuccore/MH974803 | NCBI Nucleotide, MH974803 |

The following previously published datasets were used:

| Author(s) | Year | Dataset title | Dataset URL | Database and Identifier |
|---|---|---|---|---|
| Ropars J, Toro KS, Noel J, Pelin A, Charron P, Farinelli L, Marton T, Krüger M, Fuchs J, Brachmann A, Corradi N | 2016 | Rhizophagus irregularis strain A1, whole genome shotgun sequencing project | https://www.ncbi.nlm.nih.gov/nuccore/LLXH00000000.1/ | NCBI Nucleotide, LLXH00000000 |
| Ropars J, Toro KS, Noel J, Pelin A, Charron P, Farinelli L, Marton T, Krüger M, Fuchs J, Brachmann A, Corradi N | 2016 | Rhizophagus irregularis strain A4, whole genome shotgun sequencing project | https://www.ncbi.nlm.nih.gov/nuccore/LLXI00000000.1/ | NCBI Nucleotide, LLXI00000000 |
| Ropars J, Toro KS, Noel J, Pelin A, Charron P, Farinelli L, Marton T, Krüger M, Fuchs J, Brachmann A, Corradi N | 2016 | Rhizophagus irregularis strain B3, whole genome shotgun sequencing project | https://www.ncbi.nlm.nih.gov/nuccore/LLXK00000000.1/ | NCBI Nucleotide, LLXJ00000000 |
| Ropars J, Toro KS, Noel J, Pelin A, Charron P, Farinelli L, Marton T, Krüger M, Fuchs J, Brachmann A, Corradi N | 2016 | Rhizophagus irregularis strain C2, whole genome shotgun sequencing project | https://www.ncbi.nlm.nih.gov/nuccore/LLXL00000000.1/ | NCBI Nucleotide, LLXK00000000 |
| Ropars J, Toro KS, Noel J, Pelin A, Charron P, Farinelli L, Marton T, Krüger | 2016 | Rhizophagus irregularis strain A5, whole genome shotgun sequencing project | https://www.ncbi.nlm.nih.gov/nuccore/LLXM00000000.1/ | NCBI Nucleotide, LLXL00000000 |

M, Fuchs J, Brach-
mann A, Corradi N

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
