## [Decision Letter]

Thank you for submitting your article "Single nucleus sequencing reveals evidence of inter-nucleus recombination in arbuscular mycorrhizal fungi" for consideration by *eLife*. Your article has been reviewed by three peer reviewers, and the evaluation has been overseen a Reviewing Editor, and Patricia Wittkopp as the Senior Editor. The following individuals involved in review of your submission have agreed to reveal their identity: Rene Geurts (Reviewer #1); Peter Young (Reviewer #2); Michael Seidl (Reviewer #3).

The reviewers have discussed the reviews with one another and the Reviewing Editor has drafted this decision to help you prepare a revised submission.

Summary:

Arbuscular mycorrhizal fungi (AMF) in the Glomeromycotina are ecologically important symbionts which promote plant growth and diversity. The AMF mycelium is unique among eukaryotes as it harbors hundreds to thousands of nuclei within one continuous cytoplasm and mechanisms that could trigger sexual processes in these organisms have long been elusive. This manuscript describes an elegant study to provide an answer to a long-standing question in AMF biology. Can AM fungi experience genetic recombination, despite the lack of obvious phases of sexual reproduction? Chen et al. tackled this question by assessing the hypothesis that karyogamy and genetic recombination are taking place among nuclei co-existing in dikaryotic isolates by comparing sequence data obtained from single AMF nuclei. They quantified SNP segregation in dikaryotic and homokaryotic *Rhizophagus* strains. Whole genome sequences, as well as tens of single nuclei sequence data, were generated of a total of seven strains: five from *Rhizophagus irregularis* (3 dikaryotic and 2 homokaryotic strains), and two from the related species *R. diaphanus* and *R. cerebriforme* (both homokaryotic). SNP segregation analysis provides solid evidence that genetic recombination between nuclei occurred, although at a very low rate. These findings support the contention that AMF can generate genetic diversity via inter-nuclear recombination in the dikaryotic life-stage.

The manuscript is generally well written, the illustrations and tables are clear and informative, and the applied methodology is appropriate to support the results and the conclusion, i.e. inter-nuclear recombination in AMF.

However, we have significant concerns to which extent this data and the derived conclusions present a significant step forward. Our main concerns are listed below:

Essential revisions:

1) First, evidence for recombination in different AMF have been reported previously (e.g. Croll et al., 2009, and references therein). Second, this manuscript builds on a paper from your group that was published two years ago (Ropars et al., 2016). The main new claim is evidence for recombination between nuclei in a dikaryon. In fact, Ropars et al. already claimed to have "Sporadic evidence for recombination" in their Figure 3B, though we don't see any evidence for recombination in the dikaryotic strain A5. The only putative 'recombination event' was in the homokaryotic A1, which is stated to have 'no recombination' in the current manuscript. Data as presented in Figure 1B for isolates A1, A4 and A5 have been reported previously (see Figure 1 in Ropars et al., 2016). It appears that the present paper is an extension of your seminal observations reported in Ropars et al., 2016, and provides new datasets: (i) by sequencing isolate DAOM-240409, which is considerably more divergent than A4 and A5, and (ii) by analysing a 'genome-wide' set of SNPs rather than focus on few SNP positions.

Recommendations:

– Please, carefully check for inconsistencies between the present manuscript and Ropars et al., 2016.

– Please emphasize more strongly in your Introduction/Discussion the novelty of the present findings. What's new?

2) The present manuscript presents sequence data on three strains not previously studied: *Rhizophagus irregularis* DAOM-240409, *R. diaphanus* MUCL-43196 and *R. cerebriforme* DAOM-227022. There is also analysis of sequence data for strains A1, C2, A4 and A5, but it is not clear whether this includes new data or is based on the sequencing already carried out by Ropars et al., 2016.

Recommendations: Please clarify.

3) The focus is on the dikaryotic strain DAOM-240409, and you claimed that there has been extensive recombination between co-existing nuclei in this strain, in contrast to A4 and A5, which are also dikaryotic but show only limited signs of recombination. This is certainly backed up by the similarity matrices in Figure 2. The sequencing of PCR products (Supplementary Figure 1) is less convincing, since template switching during PCR can often generate artefactual 'recombinant' sequences. The single-nucleus sequences provide much better evidence (Figure 3).

Recommendations: Please clarify and comment.

4) One important caveat is the quality of the assembly of DAOM-240409. Despite higher sequencing coverage, this strain has twice as many scaffolds as the other dikaryons, and a much larger assembly size. Although you mention this (Introduction), you offer no explanation. There may be a connection between the assembly and the high frequency of apparent recombination, but it is not clear whether the high scaffold number is somehow a consequence of recombination, or the apparent recombination is an artefact of an assembly that is poor for technical reasons. We need a more careful consideration of this question, based on some analysis of the reasons for the high scaffold number and assembly size. This could reveal new information of real biological interest. For example, is this related to any difference in transposable element density/activity/categories? There is no mention of the gene annotations in the present manuscript. Can you assess the quality of the DAOM-240409 assembly by looking at this gene annotation?

5) Reviewer #3 emphasized that the manuscript would significantly benefit from analyses aimed to provide a better understanding into the processes underlying and modulating the observed recombination (as suggested in the Discussion). For instance, while you suggest the operation of a 'meiotic-like' process, mainly due to the presence of a complete set of meiosis-related genes (Halary et al., 2011) and the absence of aneuploid nuclei, no further experimental evidence is provided. They suggested that alternative genome sequencing and assemblies (e.g. based on Oxford Nanopore or PacBio data) approaches should yield contiguous genome assemblies which should enable the analyses of recombination hotspots (Supplementary file 3) in high resolution. I assume that these additional genome sequencing cannot be carried out in time for the resubmission owing to the difficulty in obtaining high quality genome DNA from AMF for PacBio or Nanopore sequencing. Please comment.

---

## [Author Response]

Essential revisions:1) First, evidence for recombination in different AMF have been reported previously (e.g. Croll et al., 2009, and references therein).

It is essential to distinguish between two distinct patterns of recombination:

A) Past studies = PCR/Cloning based evidence for recombination among conspecific AMF isolates – e.g. Croll and Sanders, 2009, Den Bakker et al., 2010, Riley et al., 2014 and others.

B) Our study = Direct, single nucleus based evidence recombination within the AMF mycelium – i.e.how inter-nucleus recombination generates new genetic diversity in the absence of observable mating.

The origin/nature of the recombination reported by references mentioned in point (A) is very different from the one we identified in this paper. In particular, as opposed to the studies cited above, our study provide direct, single nucleus based quantifiable evidence of inter-nuclear recombination, while other studies citing evidence of recombination had to *infer its existence* based on patterns found in distant organisms.

More importantly, the above mentioned studies are all based on data obtained through PCR and bacterial cloningof single gene sequences, assuming a priori that the genes analyzed and compared among isolates were all single-copy and orthologues. Unfortunately, AMF genome needed data to back such claims was unavailable when these studies were first published.

Consequently, such claims of recombination can all be explained by a myriad of mechanisms including for example, gene conversion through paralogy (which is extensive in AMF), balancing selection, or accelerated mutation rate at the loci in some of the loci/strains investigated and, perhaps most importantly, PCR/Cloning artefacts. Indeed, to our knowledge, the only other studies suggesting that recombination may derive from inter-nuclear recombination within the AMF mycelium are those of Kuhn et al., 2001, and Galdolfi et al., 2003. Both studies are now known to have likely analyzed artificial recombinants resulting from PCR cloning procedures. Specifically, both studies identified recombination in a locus named the “Bip gene” (an Hsp90 protein), which was later found to be multi-copy and monomorphic using AMF genome data (Tisserant et al., 2013; Chen et al, 2018) as well as single nuclear genome sequences (Lin et al., 2014).

Some issues with previous claims of recombination and high heterokaryosis in AMF have been recently discussed in Corradi and Brachmann, 2017. We believe it would be confusing to resurrect these settled debates, which is the reason why we avoided discussing this particular topic in the original version of our paper.

However, we understand the editor’s and reviewer’s concerns, and we thus now briefly discuss some of the past work on AMF inter-isolate recombination in our revised paper, stressing how our study finally provides an answer to a long-lasting debate regarding the presence of inter-nuclear recombination within the AMF mycelium, particularly in the dikaryotic life-stage.

Second, this manuscript builds on a paper from your group that was published two years ago (Ropars et al., 2016). The main new claim is evidence for recombination between nuclei in a dikaryon. In fact, Ropars et al. already claimed to have "Sporadic evidence for recombination" in their Figure 3B, though we don't see any evidence for recombination in the dikaryotic strain A5. The only putative 'recombination event' was in the homokaryotic A1, which is stated to have 'no recombination' in the current manuscript.

We are sorry for the confusion the term has created. In our view, the term “sporadic” in Figure 3B legend of Ropars et al. 2016 was synonym of “isolated”. It specifically referred to the single mutation found in the homokaryotic isolate A1, which you mentioned.

In this case, the mutation could either represent a nucleus-specific somatic mutation, or a past inter-isolate recombination with the isolate C2. This situation is reminiscent of the studies mentioned earlier, where one observation (i.e. the A1 singleton) can be explained in very different ways.

In stark contrast to this, our analyses of 80+ single nuclei from seven strains provide for the first time direct, genome-wide evidence that AMF can generate genetic diversity within their mycelium by recombining nuclei with opposing *MAT*-loci. This is a completely new discovery, with outstanding implication for understanding AMF biology and genetics, and their application as biofertilizers.

Data as presented in Figure 1B for isolates A1, A4 and A5 have been reported previously (see Figure 1 in Ropars et al., 2016). It appears that the present paper is an extension of your seminal observations reported in Ropars et al., 2016, and provides new datasets: (i) by sequencing isolate DAOM-240409, which is considerably more divergent than A4 and A5, and (ii) by analysing a 'genome-wide' set of SNPs rather than focus on few SNP positions.

In our original version of the paper, we specifically state that our findings build on models proposed by Ropars et al., 2016 and Corradi and Brachmann, 2017. Specifically, we finally provide direct evidence that nuclei recombine in dikaryotic AMF isolates – i.e. AMF have found ways to generate genetic diversity in the absence of sexual reproduction. Previously, this was only a testable hypothesis.

Again, this is something that was previously unknown to exist in AMF and represents a landmark finding in the field.

More generally, the new data we provide is as follows:

– 3 new genomes assemblies: *Rhizophagus irregularis* DAOM-240409, *R. diaphanus* and *R. cerebriforme*. We would like to point out that DAOM-240409 is not phylogenetically “divergent”, but simply shows evidence of higher recombination and heterokaryosis compared to relatives. All assemblies used in this study have now been deposited in Genbank. Accession numbers are available in the revised version of the manuscript.

– Genome data from 80+ single nuclei isolated from seven *Rhizophagus* isolates, including nuclei from the *R. irregularis* isolates A1, A4, A5 and C2. A prior study provided data and analyses from 7 single nuclei (Lin et al., 2014) isolated from the model AMF *R. irregularis* DAOM-297198. Most of these analyses centered on genome content and on challenging the presence of heterokaryosis in AMF, but found no evidence of recombination in this homokaryotic isolate.

– In contrast, our analyses of single nuclei from many homo/dikryotic strains provide the first direct evidence that co-existing nuclei with opposing *MAT*-loci recombine within the AMF mycelium – i.e. we demonstrate how these organisms can create genetic diversity in the absence of observable sex.

– Demonstrate that false SNP positives are very abundant in analyses of genomes and single nuclei, and identify ways to drastically reduce their identification. This is particularly important for our area of research, which is interested in detecting the degree of genetic variability among/within isolates.

– We provide first direct (single nucleus sequencing) evidence that nuclear polymorphisms is always low in AMF, regardless of the species investigated. Previous knowledge in this area has always been inferred from single-gene or whole-genome datasets, which is always error-prone and gives a very approximate measure of genetic diversity – e.g. SNP/Kb counts.

Overall, the only publicly available data we used is represented by the genome assemblies of A4, A5, A1 and C2, which were used to map their respective single nuclei reads. All single nuclei from these isolates were newly obtained in our present study.

In the revised version of the paper, the Table 1 has been modified to highlight that the assemblies from A4, A5, A1 and C2 were previously obtained by Ropars et al., 2016.

[…]2) The present manuscript presents sequence data on three strains not previously studied: Rhizophagus irregularis DAOM-240409, R. diaphanus MUCL-43196 and R. cerebriforme DAOM-227022. There is also analysis of sequence data for strains A1, C2, A4 and A5, but it is not clear whether this includes new data or is based on the sequencing already carried out by Ropars et al., 2016.Recommendations: Please clarify.

The genome analyses presented in Figure 1 – i.e. allele plots – are based on new analyses of publicly available genome sequence data from Ropars et al., 2016 (assembly and genome reads), and new references we provide for the isolates DAOM-240409, *R. diaphanus* MUCL-43196 and *R. cerebriforme* DAOM-227022.

For clarity, this information has now been included in the revised figure legend.

3) The focus is on the dikaryotic strain DAOM-240409, and you claimed that there has been extensive recombination between co-existing nuclei in this strain, in contrast to A4 and A5, which are also dikaryotic but show only limited signs of recombination. This is certainly backed up by the similarity matrices in Figure 2. The sequencing of PCR products (Supplementary Figure 1) is less convincing, since template switching during PCR can often generate artefactual 'recombinant' sequences. The single-nucleus sequences provide much better evidence (Figure 3).Recommendations: Please clarify and comment.

We were originally surprised to see how well the recombination patterns correlated with our findings – i.e. DAOM-240409 and A4 showed evidence of recombination and found none in A5 – and thus decided to keep those results in the paper.

However, we agree that PCR//Cloning can artifactually produce these patterns (as we noted in the original manuscript) and that these could detract the reader from the solid evidence of inter-nuclear recombination we identified using single nuclei sequencing.

Thus, we decided to remove this figure from the revised version of the manuscript.

We have now modified the text accordingly. The modified section now reads as follows: “Here, we tested the hypothesis that karyogamy and inter-nuclear recombination occur in the AMF mycelium using a single-nucleus sequencing approach. […] Genome data from *R. irregularis* SL1 (also known as DAOM-240409), *R. diaphanus* and *R. cerebriforme* represent new additions to public databases from representatives of the Glomeromycotina, and all sequencing data were assembled using SPAdes (Bankevich et al., 2012) to facilitate comparisons with published genome data.”

4) One important caveat is the quality of the assembly of DAOM-240409. Despite higher sequencing coverage, this strain has twice as many scaffolds as the other dikaryons, and a much larger assembly size. Although you mention this (Introduction), you offer no explanation. There may be a connection between the assembly and the high frequency of apparent recombination, but it is not clear whether the high scaffold number is somehow a consequence of recombination, or the apparent recombination is an artefact of an assembly that is poor for technical reasons. We need a more careful consideration of this question, based on some analysis of the reasons for the high scaffold number and assembly size. This could reveal new information of real biological interest. For example, is this related to any difference in transposable element density/activity/categories? There is no mention of the gene annotations in the present manuscript. Can you assess the quality of the DAOM-240409 assembly by looking at this gene annotation?

We agree that assessing the assembly quality of DAOM-240409 is essential for this study, especially given the higher recombination rates we found in this study.

We aimed to address these concerns by performing additional analyses that follow-up on suggestions from reviewer #3. These analyses are aimed to test the completeness of the assembly and if technical/analyses errors could have generated some of the higher genetic diversity identified in the DAOM-240409 assembly.

– A BUSCO analysis was carried out on all assemblies to obtain a general overview of their completeness/coding capacity. We find that all *R. irregularis* dikaryotic assemblies almost identical in term of completeness – i.e. 90% for A4, 91% for DAOM-240409 and 92% for A5. Please note that those values are slightly lower than those reported in Chen et al., 2018, because they were obtained on genome assemblies, as opposed to annotated protein sets.

– As proposed by the reviewer, we compared TE abundance in DAOM-240409 with other isolates, and found that all show similar TE content. Thus, the higher fragmentation of the DAOM-240409 does not seem to result from higher TE counts.

– We tested whether the higher assembly fragmentation in DAOM-240409 was due to much higher genome size using K-mer counts, an approach that gives an approximate estimation of genome size based on sequencing reads. We find that all *Rhizophagus* isolates similar genome sizes – ranging between 130-150Mb.

– Higher assembly sizes and fragmentation are also found in the other dikaryons, particularly A4 but also A5, so fragmentation is clearly correlated with nuclear complexity.

For all these reasons, we believe that the larger size and genome fragmentation of the DAOM-240409 assembly is better explained by higher recombination/heterokaryosis rates, although we cannot conclusively rule out other scenarios.

These additional analyses have now been included in the revised version of the paper to better highlight our hypothesis that the higher DAOM-240409 fragmentation is the result of higher nuclear genetic diversity – i.e. recombination/heterokaryosis.

The text now highlights this and reads as follows: “Higher fragmentation is also seen in other dikaryons, particularly A4, and thus appears to result from the presence of higher intra-mycelial genetic diversity in this part of the AMF cycle – i.e. presence of two distinct nucleotypes challenges the assembly procedures. Interestingly, BUSCO, K-mer graphs and transposable elements (TE) analyses all show that SL1 harbors a gene repertoire, a genome size and TE counts that are very similar to those of other isolates analysed in this study (Table 2, Supplementary file 1)”.

Regardless of what has caused of the assembly fragmentation, it is noteworthy that we took drastic steps to reduce the detection of false positives by applying very stringent methodologies on the most reliable scaffolds – i.e. top 100 only. In summary, we have been hyper-conservative in our analyses.

5) Reviewer #3 emphasized that the manuscript would significantly benefit from analyses aimed to provide a better understanding into the processes underlying and modulating the observed recombination (as suggested in the Discussion). For instance, while you suggest the operation of a 'meiotic-like' process, mainly due to the presence of a complete set of meiosis-related genes (Halary et al., 2011) and the absence of aneuploid nuclei, no further experimental evidence is provided. He/she suggested that alternative genome sequencing and assemblies (e.g. based on Oxford Nanopore or PacBio data) approaches should yield contiguous genome assemblies which should enable the analyses of recombination hotspots (Supplementary file 3) in high resolution. I assume that these additional genome sequencing cannot be carried out in time for the resubmission owing to the difficulty in obtaining high quality genome DNA from AMF for PacBio or Nanopore sequencing. Please comment.

Our response to these comments are separated into two categories.

A) Regarding the absence of further experimental evidence for meiosis vs. parasexuality:

As discussed in the original manuscript, the observed inter-nuclear recombination can result from two processes – i.e. meiosis or parasexuality. Our goal is to always discuss all plausible scenarios for the patterns we observed.

In the original paper, we argue that meiosis better explained our findings for three main reasons. We still believe that these arguments are sound.

In particular:

– We find that a complete set of meiosis genes is found in all isolates investigated.

– Past evidence of inter-isolates recombination in AMF support the existence of conventional mating as seen in sexual fungi. Specifically, heterokaryon incompatibility prevents nuclear mixing in natural fungal populations, making it unlikely that inter-isolate recombination in AMF is not driven by mating processes and subsequent meiotic-like events.

– To date, only cytometry evidence for haploid nuclei was found.

– Parasexual cycles can drive recombination via mitotic events. However, these require the stable division of diploid cells for a number of generation to trigger mitotic recombination, but we found no sequence-based evidence for diploidy in our single nucleus analyses.

– The high frequency of recombination found in DAOM-240409 is not compatible with a parasexual cycle, as recombination generally happens at low frequency during this process.

The paper was revised to discuss this additional information. The modified text reads are follows: "In fungi, diploid nuclei can undergo recombination either through meiosis, in a sexual cycle, or through aneuploidization, in parasexual cycle involving nuclear fusion followed by random chromosomal loss. […] For these reasons, we argue that the inter-nuclear recombination we observed in this study is likely driven by meiotic events – i.e. conventional fungal sexual cryptic process – although it is possible that parasexuality has also been at play in creating some of the observed nuclear diversity…”

As original stated, we still fully acknowledge the potential contribution of parasexuality in creating some of the observed recombination.

B) Regarding long-read sequencing:

We agree with the referee that having access to “long-reads” sequencing data would be very helpful in this study. Our long-term goal is to acquire optimal, ideally chromosome-level, assemblies using such technologies. To this end, we are presently trying to acquire large quantities of High Molecular Weight DNA from all AMF dikaryotic isolates, but this is has been more challenging than we had anticipated.

Consequently, as recognized by the editor, it will take us well over two months to implement optimal conditions for AMF DNA extractions, obtain top assemblies, and analyze our single nuclear data using these technologies. Nevertheless, we understand the concern of the reviewer, and thus we aimed to address it using two independent approaches. Both of these methods confirm our previous findings:

– We first tested if a better assembly for DAOM-240409 would remove evidence of recombination in this isolate. To this end, we used ALLPATHS-LG and obtained a significantly better assembly for SL1 – i.e. 6642 scaffolds, as opposed to 29249 for the SPAdes assembly. We find that using identical methods on this new assembly still shows substantial evidence of recombination in this isolate.

Furthermore, single nuclei pairwise genotype similarities based on the ALLPATH-LG assembly are very similar to those found based on the SPAdes assembly.

– Secondly, we tested if recombination could also be found using known stringent “variant callers” such as GATK HaplotypeCaller and Mutect2. Both callers again supported the existence of significant inter-nucleus recombination in DAOM-240409.

These additional analyses have now been included in the revised version of the manuscript and are discussed, accordingly in the paper.

The revised text reads as follows: “Performing identical analyses on improved assemblies (ALLPATHS-LG; Butler et al., 2008; Supplementary file 7) and with more stringent variant callers (GATK-HaplotypeCaller (Van der Auwera et al., 2002), Mutect2 (Cibulskis et al., 2013); Supplementary files 8 and 9) did not change the overall findings. […] Accordingly, the genetic relationships among co-existing nuclei are also generally unaffected by changes in both assembly and SNP scoring methods (Supplementary files 10-12)”.

Accordingly, several new supplementary files were added to the revised manuscript.